# Strategies for Sudden Cardiac Death Prevention

**DOI:** 10.3390/biomedicines10030639

**Published:** 2022-03-10

**Authors:** Mattia Corianò, Francesco Tona

**Affiliations:** Department of Cardiac, Thoracic, Vascular Sciences and Public Health, University of Padova, 35128 Padova, Italy; mattia.coriano@studenti.unipd.it

**Keywords:** sudden cardiac death, risk stratification, cardiovascular magnetic resonance, neural network, machine learning, artificial intelligence, cardiomyopathy

## Abstract

Sudden cardiac death (SCD) represents a major challenge in modern medicine. The prevention of SCD orbits on two levels, the general population level and individual level. Much research has been done with the aim to improve risk stratification of SCD, although no radical changes in evidence and in therapeutic strategy have been achieved. Artificial intelligence (AI), and in particular machine learning (ML) models, represent novel technologic tools that promise to improve predictive ability of fatal arrhythmic events. In this review, firstly, we analyzed the electrophysiological basis and the major clues of SCD prevention at population and individual level; secondly, we reviewed the main research where ML models were used for risk stratification in other field of cardiology, suggesting its potentiality in the field of SCD prevention.

## 1. Introduction

Sudden cardiac death (SCD) represents a major challenge of modern medicine. The prevention of SCD orbits on two levels, the general population level and individual level. However, current recommendations are full of gaps in evidence, and the strategies proposed do not fit a large number of patients. In this scenario, this review has two aims: first, to introduce the electrophysiological basis of SCD and to evaluate the state of the art on SCD risk prevention and its weaknesses; second, to propose a new field of research based on artificial intelligence (AI) for SCD prevention. These purposes will be analyzed before focusing on a population level and then on an individual level. 

SCD is defined as an unexpected death not attributable to an extracardiac cause, usually within 1 h of symptom onset, or within 24 h of last being seen in good health if the death is unwitnessed [1]. Acute coronary syndrome is the leading cause of SCD, and other frequent causes are cardiomyopathies, myocarditis and inherited channelopathies. The electrophysiological basis concern about a state of electrical instability due to alteration in ion channels currents. SCD accounts for up to 50% of all cardiac deaths, and 50% of SCDs represent the first cardiac event [2]. This suggest that SCD prevention through a high accuracy risk prediction should be achieved by precision medicine. Risk of SCD could be described considering two different epidemiologic levels. First, the general population-based level focuses on large groups of patients without a prior history of heart disease, and aims to identify individuals who could benefit from a more intensive control of risk factors and a closer follow up. Second, the individual-based level focuses on a smaller group of patients, affected by a specific heart disease, with the aim to identify who could benefit from an implantable cardiac defibrillator (ICD) for primary prevention. Much research has been done with the aim to improve risk stratification of SCD, although no radical changes in evidence and, therefore, in therapeutic strategy have been achieved. A novel approach for future research is represented by the use of machine learning (ML) technologies to improve risk prediction of SCD. ML is considered a subgroup of the artificial intelligence (AI) based on building models that make accurate predictions from input data. Models are based on algorithms created without needing to pre-program rules, and they can learn prediction rules automatically from the available data. Cardiovascular researchers’ interest in ML grew exponentially in the last years, with an increase of research in this field (Figure 1) [3]. About that, a bibliometric survey revealed that in the past 5 years, over 3000 papers on ML in cardiology were published in PubMed. The majority of publications involve computed tomography for atherosclerosis diagnosis, ECG finding and relation with arrythmias, electronics heart record or ultrasound for heart failure (HF) diagnosis and prognosis [4]. Despite this elevated number of publications, the subject of SCD lacks original research in risk prediction using an ML model.

## 2. Electrophysiological Basis of Sudden Cardiac Death

### 2.1. Cardiac Action Potential

To understand the pathophysiology of SCD, it is necessary to briefly review the electrophysiological basis of cardiac action potential (AP) (Figure 2). When a myocardial cell receives the impulse from the conduction system, a series of mechanism are activated, which leads to the following events: firstly, the opening of Na^+^ voltage-dependent channels determines an inward current on Na^+^ (I_Na_), resulting in a rapid depolarization of the cell (phase 0). Secondly, an early repolarization results from the opening of a specific subgroup of K^+^ voltage-dependent channels, which determines an outward current of K^+^ (I_Kto_) that reduce the electric potentials of the cell (phase 1). Thirdly, a plateau phase (phase 2) of the electric potential is due to a balance between an inward current of Ca^2+^ (I_Ca-L_), mediated by L-type Ca^2+^ voltage-dependent channels, and an outward current of K^+^ mediated by two types of channels (respectively, K_r_ in the first period and K_s_ in the second period of the plateau phase). During this phase, there is a concomitant flow of Ca^2+^ from the sarcoplasmic reticulum to the cytoplasm of myocardial cells, allowing the excitation-contraction coupling resulting in the contraction of the cell. Fourthly, an outward current of K^+^, mediated by K_s_ and K_1_ voltage-dependent channels, determines the late repolarization (phase 3), which restores the basal electric potential. Fifthly, a balance between K^+^ outward current, mediated by K_1_ voltage-dependent channels (I_K1_), and the activity of Na^+^/K^+^-ATPasi restores the ionic equilibrium at rest (phase 4) and prepares the cell for a new depolarization phase [5].

### 2.2. Basis of Ventricular Arrhythmias

SCD is frequently caused by a cardiac arrhythmia, most often ventricular tachycardia (VT) or ventricular fibrillation (VF), which impairs cardiac pumping. The pathophysiology of SCD embraces different mechanism, with a variable weight of genetic and acquired factors. In a simplistic model, there are two major requisites for arrhythmic events: a vulnerable myocardial substrate and a trigger. A vulnerable myocardium represents a locus where the myocardial cells have abnormal properties of ionic currents, leading to electrical disorder and aberrant impulse generation or propagation. Heterogeneous conduction causes the emergence of spiral wavers, impulses that travel in a circular pattern around an anatomic barrier, as in a scarred myocardium or around a point of reentry. Persistent rhythmic spiral wave propagation determines VT, and its degeneration into smaller wavelets determines VF. Triggers of cardiac arrhythmias mainly consist of an ectopic excitation generated in the ventricles, causing spiral waves of depolarization in a vulnerable myocardium. Focal ectopic impulses usually append in the phase 2 of AP, named early afterdepolarization impulses, and are due to an increased activation of Ca^2+^ channels or persistent activation of Na^+^ channels, or in the phase 3–4, they are named delayed afterdepolarization impulses and are due to a spontaneous intracellular Ca^2+^ release that evokes a transient inward Na^+^ current. Prolonged AP duration increases the propensity of myocardial cells to exhibit early ectopic impulses, whereas increased Ca^2+^ loading of the sarcoplasmic reticulum predisposed to late ectopic impulses [5].

### 2.3. Vulnerable Myocardium

The different etiologies of SCD play a role in the physiopathology of ventricular arrhythmias, making the myocardium susceptible to an arrhythmic event, and this vulnerability consist of a complex interaction between inherited predisposition and acquired factors. Regarding inherited predisposition, there two categories of monogenic heart disease predisposition to SCD, resulting in disorders of heart rhythm and familial cardiomyopathy. The genes responsible for congenital arrhythmia syndromes encode predominantly either ion channel subunits or proteins that interact with ion channels (Figure 2). Overall, abnormal function of ion channels predisposed to SCD is caused by three mechanisms: abnormal repolarization, slow ventricular conduction and aberrant intracellular Ca homeostasis. Mutations in genes encoding in voltage-gated K^+^ or Na^+^ channels subunits, L-type Ca^2+^ channel, inwardly rectifying K^+^ channels and various channel-interacting proteins determines an abnormal repolarization, resulting in prolonging or shortening of QT interval on the electrocardiogram (ECG). Loss-of-function mutations in K^+^ channels or gain-of-function mutations in Ca^2+^ or Na^+^ channels increase the time of repolarization, resulting in a long QT syndrome (LQTS). Conversely, gain-of-function mutations in K^+^ channels, or loss-of-function mutations in Ca^2+^ channels, decrease the time of repolarization, resulting in a short QT syndrome (SQTS). Moreover, a slow ventricular conduction is mainly secondary to a loss-of-function mutations in Na^+^ channels, resulting in an increased time of depolarization (increased time of phase 0 of AP). This gene mutations represent the electrophysiological basis of Brugada Syndrome, where a reduction in the myocardial Na^+^ current determines a disaggregation in the AP duration between the endocardium and epicardium layers of the ventricular muscle, resulting in a substrate promoting reentrant arrhythmias. Furthermore, alterations in intracellular Ca^2+^ homeostasis represent the pathophysiological basis of catecholaminergic polymorphic VT (CPVT). Mutations often occur in RYR2 genes, encoding the cardiac ryanodine receptor/Ca^2+^ release channel, CASQ2, encoding the sarcoplasmic reticulum Ca^2+^-binding protein calsequestrin, or TRDN, encoding triadin, which links RYR2 with calsequestrin. These mutations determine an exaggerated spontaneous release of Ca^2+^ from sarcoplasmic reticulum, which are predisposed to VT when a beta-adrenergic stimulation evokes Ca^2+^ release during diastole [5].

Different considerations should be made for cardiomyopathies and HF, where different mutations at the level of contraction unit of the myocardial cells determine a wide spectrum of myocardial dysfunction, which progressively lead to HF and predisposition to SCD. The core of electrophysiological abnormalities in HF consists of a prolongation of the AP, because of a delayed repolarization secondary to a reduction in K^+^ outward current at phase 1, 2 and 3. These alterations lead to a susceptibility of the myocardium to the early afterdepolarization phenomenon, similarly to what has been seen in LQTS. Moreover, some oscillatory prepotentials, described as a series of subthreshold oscillation in resting membrane potential secondary to alteration of Na^+^/Ca^2+^ exchange and reactivation of L-type Ca^2+^ current, are able to initiate automaticity and determine spontaneous depolarization that leads to ventricular arrhythmias [6].

One last consideration deserves myocardial infarction (MI). The scar related to MI provides the underlying substrate for ventricular arrhythmias, whereas abnormalities in cardiac conduction, repolarization or autonomic modulation play the role of arrhythmic trigger. In an acute phase after MI, VT/VF can develop because of changes in cellular electric activity, particularly repolarization problems. After 5 days of MI, cells in the infarct core and the border zone have a prolonged refractory period. The resultant regions of slow conduction allow the development of re-entrant circuits with consequent tachyarrhythmia. Although the underlying scar is responsible for the development of a re-entrant circuit, neurohormonal mechanisms and LV remodeling in patients who develop HF increase the propensity to develop arrhythmia [7].

## 3. Sudden Cardiac Death in the General Population: Incidence, Risk Stratification and New Perspective

### 3.1. Lifetime Incidence 

Evaluation of real incidence of SCD in adult population is limited, because of the different epidemiological methods used in worldwide registries. In the 2020 Update of Heart Disease and Stroke Statistics report of the American Heart Association (AHA), the estimated incidence of out-of-hospital cardiac arrest assessed by emergency medical services per 100,000 population was 86.4 and 98.1, respectively, for Europe and North America [8]. Moreover, most cardiac arrests arose without warning symptoms and were always fatal [2]. These data, derived from large national and international registers, are useful to understand the magnitude of SCD in terms of morbidity and mortality, and to plan large-scale programs of intervention, but do not reveal strong information about risk factors that could permit to identify a subgroup of high-risk patients. Lifelong risk of SCD in the general population was evaluated only in one prospective cohort study. Bogle et al. [9] used individual data from the Framingham Heart Study original cohort to estimate the cumulative lifetime risk for SCD in men and women, stratified by sex, age and cardiovascular risk factor (current smoking, blood pressure, cholesterol levels and diabetes mellitus). Overall lifetime risk at index ages ranged between 10.1% and 11.2% in men, and between 2.4% and 3.4% in women, with an increase in risk, respectively, at 12% and 5.1% in those with at least two major risk factors. Relation between age and SCD was not linear, as the majority of events occurred before age 70 years, suggesting a subgroup of patients who are at risk of premature mortality that could be preventable. The nonlinear relationship between age and SCD, the sex difference and the incremental burden of cardiovascular risk factors are explained considering that the first recognized etiology of out-of-hospital SCD is ascribable to an ischemic heart disease syndrome. This assumption implies that the risk stratification in the general population should take into account the presence and severity of cardiovascular risk factors [10]. The results of this study were independent from the prior history of cardiovascular disease in the cohort population. This result agreed with older data suggesting that the majority of SCD occurred in the general population without clinically recognized heart disease, typically as the initial manifestation of cardiovascular disease [11,12,13].

### 3.2. Risk Stratification

Most research and advances in SCD risk stratification focused on an individual level. This approach is reductive because it does not consider that the majority of SCD occurs in patients with no known heart disease. This suggests that more sophisticated models are requested for risk stratification in the general population. On this field, only one study tried to build a risk score for SCD using identified markers [10,14]. Deo et al. [14] built a 10-year prediction model in a cohort of 17,884 patients, with a sample of 13,677 used as training cohort, and a sample of 4207 patients as validation cohort. The analysis excluded participants who had prevalent cardiovascular disease (history of coronary heart disease, HF or stroke). The model was built using the following variables: age, male sex, black race, current smoker, systolic blood pressure, use of antihypertensive medication, diabetes mellitus, serum potassium, serum albumin, high-density lipoprotein, estimated glomerular filtration rate and the corrected QT interval. The model was then tested in the validation cohort, and was compared to the 2013 American College of Cardiology/American Heart Association (ACC/AHA) cardiovascular disease (CVD) Pooled Cohort risk equations, which was developed to calculate the 10-year risk of a first cardiovascular event, including nonfatal MI, coronary heart disease death or fatal or nonfatal stroke [15]. The SCD prediction model resulted in moderate accuracy in risk prediction of SCD, with a C-statistic of 0.820 (95% CI 0.785–0.854), and in a slightly better performance compared with ACC/AHA CVD Pooled Cohort risk equation, which resulted in a C-statistic of 0.745 (95% CI 0.701–0.789). Rather than the slight improvement in accuracy of SCD risk prediction, this study imparts two important messages. Regarding the distribution of risk, the model was developed across deciles of risk and demonstrated a nonlinearity of risk gradients, with the major impact in the highest deciles. In particular, a 5% SCD risk over 10 years was observed in the validation cohort, and an 11% risk in the test cohort. Moreover, a subgroup of patients in this study underwent an echocardiography assessment; the majority had a normal left ventricular ejection fraction (LVEF), and this variable did not correlate with SCD.

### 3.3. New Perspectives

The underpower of actual SCD prediction models was extensively reviewed by Myerburg and Golberger [16]. According to them, a major limitation consists of using a specific cut-off of a continuous variable to separate patients at high risk rather than low risk. Some examples are the value of LVEF, with a cut-off value of 35% considered in many of ICD trials, and the use of dichotomic variables for risk factors such as diabetes rather than the glycated hemoglobin. Moreover, the regression-based techniques assume a linear relationship between risk factors and the outcome, providing in an oversimplified and approximate risk assessment. The limitations mentioned in SCD risk prediction occur either for CVD prevention. In fact, the latest ESC guidelines on CVD prevention in clinical practice suggest, for patients without diabetes and previous CVD, the use of the Systematic Coronary Risk Evaluation 2 (SCORE2) to estimate 10-year fatal and nonfatal CVD risk, in individuals in Europe aged 40–69, and SCORE2 Older People in aged ≥70 [17,18,19]. As mentioned above, these kinds of scores make an implicit assumption that each risk factor is linearly related to CVD outcomes, reducing the complex relationship between variables, and therefore, the accuracy of prediction. The need for more accuracy in risk prediction was amply analyzed in the field of SCD and CVD, and different solutions were proposed [16,20,21]. In particular, many authors evaluated the utility of ML models to improve accuracy of CVD risk prediction, but none did the same research in the field of SCD prevention. 

Table 1 shows the main studies focused on building an ML model for CVD risk prediction using cardiovascular risk factors and humoral biomarkers as input variables. Many of these studies used a different ML algorithm to predict CVD compared with the current risk stratification models suggested by guidelines. Theoretically, there is an infinite number of algorithms that could be used for prediction models. In practice, current research in healthcare focused on a subgroups of “algorithm families”: linear regression estimators, trees and neural networks (NN) [22]. Linear regression (LR) is a statistical method which describes the relationship between a qualitative dependent variable and an independent variable. Decision trees (DTs) are often used for medical decision making and resemble a flowchart, where each branch of the tree splits the study population into increasingly smaller subgroups with low within-group and high between-group variability; DTs are able to consider nonlinear relationships and have a good interpretability but are prone to overfitting. The concept of DT has been implemented, with the intuition that fitting multiple DTs improves accuracy of prediction and/or classification and reduces the overfitting. In particular, the two most common methods derived from DT are gradient boosting machine (GBM) and random forest (RF). GBM learns from the errors made by the first tree to produce a more optimal tree in the next iteration (so-called boosting strategy). RF builds a forest composed of multiple DTs fitted to a resampled version of the data with a random subset of covariates. GBM and RF have a high level of accuracy performance but do not explain the mechanism for the results [22,23,24]. NN is a series of data transformations where the outputs from one series of transformations informs the input to the next series. The architecture of an NN reflects the connection of the human brain and is stratified in different layer, in which different nodes are located (reflecting neurons of the central nervous system). Each node in a hidden layer receives input from several nodes in the prior layer, and the weight of each input is adjusted during iterative training, to produce more accurate responses. Different types of NN, reflecting the different structure of the layers, are used for different data types: for example, convolutional NN (CNN) exploits spatial dependencies among image pixels and are useful for images; recurrent NN exploits dependencies over time and is useful for language, time series and health records. With the empowering of the computational process, NNs have been used more frequently in cardiovascular medicine, both to solve classification and prediction problems; they are useful for predicting outcomes with highly complex, nonlinear relationships and interactions. The limitations of NN are the lack of explanation of how NN transform data, resulting in a “black box effect” [25,26]. Other algorithms used for CVD prediction models are briefly explained in Table 2. 

Overall, most of the studies reported in Table 1 resulted in a higher performance of ML algorithms compared with conventional guidelines-based risk stratification. It is important to underline that the gain in accuracy attended does not allow to assert a net superiority of ML models than usual regression-based methods for CVD risk prediction, except for the model built by Kakadiaris et al. Moreover, Li et al. found a similar performance between ML models and regression models. The reason lies in the source of the data used by the authors, which consist of primary care datasets (e.g., Clinical Practice Research Datalink) or prospective cohort datasets realized for epidemiologic purpose rather than for ML models building (e.g., UK Biobank), resulting in a variable quality of primary data with a high number of missing data. This remarks the concept that the accuracy of an ML model strictly depends on the quality of the input data, in particular in the training and test phase of the model building; once the model is made, it could be validated in an external cohort with a lower quality of data, considering that ML could represent a novel opportunity to improve the accuracy of risk prediction of CVD at a population-based level. Moreover, these algorithms are scalable, and the results obtained in CVD could be reproduced in the field of SCD prevention, with the aim to increase the number of patients at high risk correctly identified, who could benefit from a closer follow-up and a more intensive treatment strategy.

## 4. Sudden Cardiac Death Risk Management at an Individual Level: Current Recommendations, Gap in Evidence and New Perspective

The problem of SCD risk stratification reaches a maximum of complexity at an individual level, where patients suffer from different heart disease, and clinicians have to detect who could benefit from an ICD implantation for primary prevention. Over several decades, a multitude of studies has focused on the identification of major clinical risk markers that stratify patients according to level of risk, with the aim to identify high-risk patients who could benefit from ICD implantation. A heated debate takes place on the field of cardiomyopathy, where newer research suggests many gaps in evidence in current recommendation for SCD primary-prevention strategies in nonischemic dilated cardiomyopathy (NI-DCM), hypertrophic cardiomyopathy (HCM) and arrhythmogenic cardiomyopathy (AC). In reverse, less evidence is present in the field of channelopathies, and diagnostic algorithms proposed by current guidelines are not widely accepted.

### 4.1. Sudden Cardiac Death Prevention in Channelopathies

Risk stratification of SCD appears more complex in channelopathies than in structural heart disease, because of the relative rare incidence, the absence of persistent ECG typical aspects in a relevant percentage of patients and the extreme variability of phenotypic expression. Overall, in channelopathies, appropriate ICD therapy for VF/VT is reported in 8% to 33% of patients, while inappropriate shocks and device complications are reported in 8% to 35% [35]. 

This remarks the need for more adequate risk stratification algorithms. AHA guidelines for management of VT and SCA suggest different strategies depending on the type and subtype of the pathology. LQTS patients are defined at high-risk of SCD if they present QTc >500 ms, genotypes of LQTS type 2 and 3, are females with genotype of LQTS type 2, <40 years of age, present onset of symptoms at <10 years of age and present recurrent syncope. If these patients remain symptomatic or with QTc > 500 ms after optimization of medical therapy, they could benefit from ICD implantation. Moreover, in patients with LQTS, the finding of a phenotype-related genetic mutation is reported in 50–85%, and approximately 10–36% of genotype-positive patients with LQTS have QTc ≤440 ms [36]. In reverse, less clinical evidence subsists for risk stratification of SCD in SQTS. Patients with QTc ≤300 ms, with a history of documented polymorphic VT/VF and unexplained syncope result at increased risk of SCD, but no clear therapeutic algorithm has been well established [36]. In CPVT syndrome, the implantation of ICD should be reserved for patients with prior cardiac arrest, refractory ventricular arrhythmias on top of medical therapy or in patients who develop exercise-related symptoms. As opposed to LQTS, CPVT therapy is not guided by genotype status, because of a poor correlation between genotype and phenotype of the syndrome [36]. Finally, in Brugada Syndrome the process of decision making for ICD implantation results more complex and is under evaluation by multiple studies. According to AHA guidelines, risk stratification is based on symptoms and clinical findings, while genotype status is not correlated with the risk of adverse events. Patients with spontaneous coved type ST elevation, or drug-induced coved type ST elevation, and a history of syncope or prior cardiac arrest are at highest risk for lethal VT/VF and could benefit from ICD implantation [36].

### 4.2. Sudden Cardiac Death Prevention in Cardiomyopathy

Latest ESC guidelines recommend NI-DCM an ICD implantation for the primary prevention of SCD in patients with symptomatic HF (NYHA class II-III) and LVEF ≤ 35%, despite ≥3 months of OMT with a class IIa of recommendation [37]. These are slightly different from the AHA guidelines, which for patients with the same characteristics recommends the implantation of ICD with a class I of recommendation [36]. The rationale of this statement can be found in a recent meta-analysis that reviewed six primary prevention trials, demonstrating a 24% reduction in all-cause death with an ICD implantation [38]. Although, DANISH study, the most recent of six, showed that ICDs did not reduce overall mortality rather than OMT in NI-DCM, highlighting a gap in evidence that could change future decision making in the field of SCD prevention [39]. Moreover, data from nationwide registers showed that LVEF is an important prognostic factor in NI-DCM, but most patients who experienced SCD do not have severely reduced LVEF, and many patients with significant impairment of LVEF may still be at low risk for SCD [40]. The prospective data registry “Oregon Sudden Unexplained Death Study” examined 2093 cases of SCD in a period of about 10 years from an intake population of 1 million, showing that 68% of patients had an LVEF >35% and would have been excluded from ICD therapy under current guidelines [41]. In this scenario, many researchers investigated the role of other characteristics rather than LVEF for risk stratification of SCD, and interesting discoveries have been done with the use of cardiac magnetic resonance (CMR). In particular, the analysis of late gadolinium enhancement (LGE) to evaluate the presence of midwall fibrosis showed a correlation between arrhythmic events and areas of LGE [42,43,44,45,46,47,48,49]. No clinical trials have investigated the superiority of a CMR-guided strategy for ICD implantation, although one trial is currently ongoing, testing the hypothesis that “in patients with mild to moderate left ventricular dysfunction and CMR evidence of myocardial fibrosis, a strategy of ICD insertion will be superior to standard care” [50,51,52,53].

Different consideration should be made for HCM, where risk stratification and selection of patients continue to evolve. In 2014, O’Mahonny et al. realized a score model for SCD prediction in HCM (HCM-Risk SCD) derived from five cohort for a total of 3675 patients, including following predictor variables: age at time of evaluation, family history of SCD in ≥1 first-degree relatives aged <40 years or a first-degree relative with confirmed HCM at any age, ventricular wall thickness at echocardiography, left atrial diameter, maximal left ventricular outflow tract gradients, presence of nonsustained VT on Holter monitoring and unexplained syncope. HCM-Risk SCD was based on regression methods and permit to stratify patients with HCM in three risk categories, the high-risk category with estimated 5-year risk ≥ 6%, the low-risk category with a 5-year risk <4%, and the intermediate-risk category with a 5-year risk of 4–5%. The score resulted in an acceptable performance at internal validation, with a calibration slope of 0.876 (95% CI: 0.869, 0.883) and a C-statistic of 0.69 (95%: CI 0.68,0.71), and at external validation in a cohort of 3703 patients with a calibration slope of 1.02 (95% CI, 0.93–1.12) and a C-statistic of 0.70 (95% CI, 0.68–0.72) [54,55]. The efficacy of HCM-Risk SCD was reiterated in a meta-analysis comprehensive of six studies in an overall population of more than 7000 patients, resulting in a high accuracy in prediction of SCD in low-risk and high-risk patients, with a C-statistic ranged from 0.69 to 0.92. Subsequent research developed new imaging features of HMC related with risk of SCD, not included in HCM-Risk SCD. In particular, the presence of a left ventricular apical aneurysm, LGE and LVEF <50% are associated with an increased risk of SCD, highlighting a partial incompleteness of HCM-Risk SCD [56,57].

Similar consideration concerns AC. Both European and American recommendation suggest to follow a shared decision-making algorithm for prophylactic ICD implantation that takes into account of the value of LVEF, the male sex and the presence of nonsustained VT, specific genic mutations (in particular phospholamban, laminin A/C and filamin-C), right ventricular dysfunction and proband status [58]. Recently, Cadrin-Tourigny et al. [59] developed a prediction model based on a cohort of 528 patients to predict the incident sustained ventricular arrythmias considering the following predictor variables: sex, age, recent (<6 months) cardiac syncope, nonsustained VTs, number of premature ventricular complexes on 24 h Holter monitoring, extent of T-wave inversion on anterior and inferior leads, right ventricular ejection fraction and LVEF. The model was assessed using regression methods, and resulted in an acceptable prediction performance, with a C-statistic of 0.77 (95% CI 0.73–0.81) and a calibration slope of 0.93 (95% CI 0.92–0.95). Although current recommendations do not consider CMR as a primary tool in decision-making for ICD implantation, recent research enhanced the prognostic role of LGE in AC, as a predictor of VT (relative risk 3.13 [95% CI 1.69–5.78]) and ICD implantation (relative risk 3.15 [95% CI 1.69–5.87]) [60,61,62]. Despite what has been mentioned above, current European recommendations suggest to evaluate the decision of ICD implantation for primary prevention based on the HCM score. On the other hand, AHA/ACC guidelines for SCD prevention do not recommend the use of the HCM score for a decision-making process, rather suggest to implant an ICD for primary prevention in patients who present at least one of the following risk factors: maximum LV wall thickness ≥30 mm, family history of SCD or any episode of unexplained syncope within the preceding 6 months [36,37]. 

Such consideration enhanced some critical issues. First, current recommendations of CMD are based on results of old clinical trials, where recent evidence suggests that different aspect rather than LVEF are more appropriate for decision making of ICD implantation. Second, HCM risk and the prediction model built for AC and HCM are assessed using regression methods, and therefore, implicitly share the problems of approximation and simplification of risk assessment, as explained in the first part of this review. Third, current recommendations do not emphasize the importance of advanced imaging technique, especially CMR and LGE analysis, in the decision-making process of ICD implantation. This aspect represents the greatest potential field for future research and clinical trials, because of the high accuracy of CMR to identify endomyocardial fibrosis, which acts as a substrate for tachyarrhythmias. Pioneering research was carried out by Gulati et al. [42], analyzing the association of LGE, as a surrogate of midwall fibrosis, with SCD in patients with NI-DCM. They found that addition of LGE presence at LVEF resulted in a net reclassification improvement of 0.29 for arrhythmic composite end point. In the following years, much other research was done in the subject of cardiomyopathies, and the prognostic value of LGE for arrhythmic event was confirmed by a recent metanalysis for NI-DCM and HCM or AC [57,62,63,64].

### 4.3. Image Analysis and Role of Neural Network

In the last decade, the spread of new technologies, such as more computational power and more capacious systems of data storage, allowed the development of ML algorithms to perform imaging analyses. In cardiovascular imaging, and in particular in CMR, ML models have been tested for three different purposes: automation, diagnosis and prediction [65] (Figure 3). Automation models tried to reduce time spent by an operator for the acquisition of CMR images [66,67,68] and segmentation of chambers and myocardium [69]. Diagnostic algorithms were built to improve accuracy in the detection of tissue features and clustering features pattern. The potentiality of ML models to extract a high number of image features facilitate the growth of “radiomics”, a novel subject with the scope of converting digital medical images into mineable high-dimensional data. Various features can be extracted from images, as morphologic, intensity-based, fractal-based and texture features [70]. In particular, texture analysis (TA) permits to investigate the distribution and relationship of pixel or voxel gray levels in the images, and is increasingly proving to be a fertile subject for ML application [71]. TA based on ML models was performed to detect differences in LGE pattern in the patients with Tako-Tsubo syndrome, HCM and HF with systolic dysfunction. The features identified resulted in an improvement in diagnostic performance and the differences found reflected differences in prognostic outcomes [72,73,74,75,76]. Furthermore, image analysis with ML was used to predict outcome in patients with CVD. Kotu et al. [77] used k-nearest neighbor (k-NN), support vector machine (SVM) and random forest to predict the occurrence of cardiac arrhythmia after MI (Table 2). They found that CMR-derived scar texture features based on scar gradient demonstrated a risk stratification power comparable to currently used criteria, such as LVEF and scar size in discriminating patients at high risk of arrhythmic event from patients at low risk. In particular, the SVM model was the more accurate model, showing a maximum of C-statistic of 0.97 (95% CI 0.85–0.98). Moreover, Bello et al. [78] used CMR features together with clinical information to train a CNN-based classifier that can predict outcome in patients with pulmonary hypertension. The model demonstrated a predictive accuracy with a C-statistic of 0.75 (95% CI 0.70–0.79) in discriminating patients at high or low risk of cardiovascular death.

### 4.4. Future Perspectives

The progress presented above shows the work already done in risk prediction of SCD, and future research could offer new alternatives for SCD prevention in cardiomyopathies. As mentioned, recent evidence in CMD, HMC and AC suggested a prognostic role of CMR features, whose potential is not fully exploited. This is important because human accuracy in analyzing an enormous quantity of data, such as CMR images, is low compared with the accuracy of ML algorithms. In particular, CNN is very useful for this purpose. These kinds of algorithms use a convolutional process that mimics how the visual cortex processes images. In a CNN, an image is separated into components, and convolutions are used to identify local correlations between input data. Convolutions permit a neuron to receive input from only a subset of nodes in the prior layer, preserving only local and spatial relationships [79]. Finally, NN allows to integrate images to row data as clinical data on risk factors, symptoms or genetic phenotype, resulting in the possibility to take into consideration a major number of characteristics that previous research related to SCD has shown, but which is not fully considered in current guidelines. Although this possibility appears tempting, a relevant reflection should be done. ML, and in particular NN, have the potential to include a very high number of variables, if a correlation between variables and clinical endpoint was not previous demonstrated. That would imply an incorrect estimation of the weight of each variable and will never produce a true evidence-based risk stratification process. Regarding this, our statement is that every variable included in an ML model should have a proven correlation with the event that we are trying to predict. 

## 5. Conclusions

SCD prevention remains a major challenge of modern medicine. The heterogeneity of its physiopathology, where acquired and genetic factors concur to the development of fatal ventricular arrhythmias, affect the strategies of risk prevention. As shown in Figure 4, in this review, we evaluated risk stratification in two different dimensions, population level and individual level. Current evidence reveals many problems in the accuracy of risk stratification strategies at each dimension level. Solutions proposed in this review suggest the use of ML models to improve patient selection for an ICD implantation for primary prevention. At the moment, the cardiology community looks at ML ambivalently. On the one hand, the possibility of progress in direction of precision medicine generates enthusiasm, established by the growing number of experimental research on this subject. On the other hand, some negative criticism has been made. One major criticism is the inability to interpret the output of ML models, creating the so-called “black box effect”. Fortunately, several tools are being developed to improve the explanation of ML results [80,81]. Another criticism is the need for a high number of well-prepared data to train algorithms, implying a high amount of time spent by clinicians to build these kinds of datasets. Finally, an ethical regulation is required, in particular concerning data sharing, data privacy and security testing of models. In this field, both the USA and Europe launched initiatives on AI to ensure an appropriate ethical and legal framework [82].

## Figures and Tables

**Figure 1 biomedicines-10-00639-f001:**
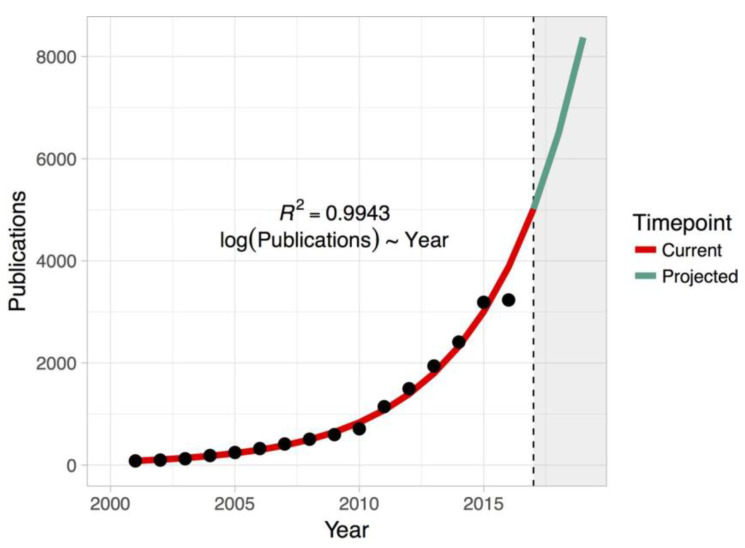
Projecting the growth of publications in PubMed ‘machine learning’. Exponentiated regression of log number of publications on year is used to predict the future trend (adapted from Shameer et al. [3]).

**Figure 2 biomedicines-10-00639-f002:**
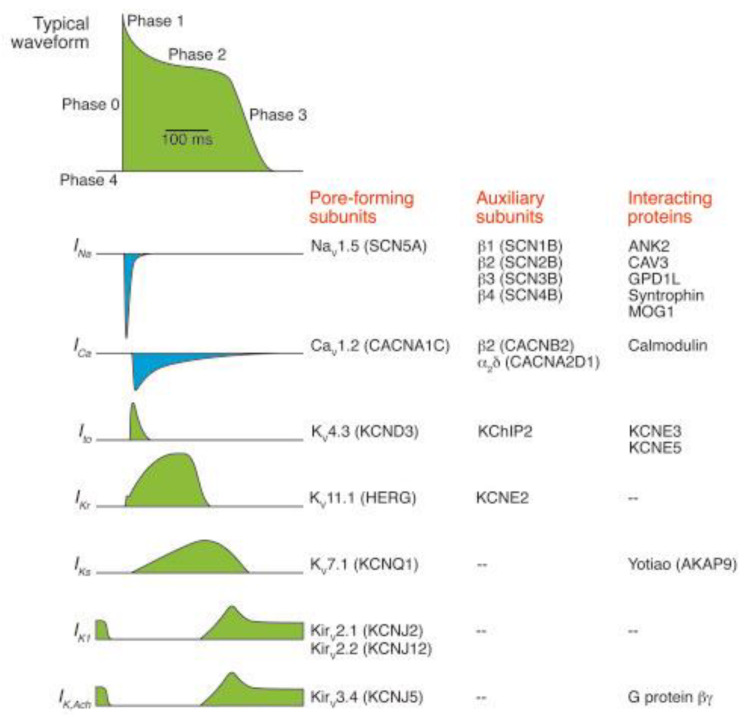
Ionic and molecular basis of cardiac action potential. Left: the ventricular action potential waveform with different phases and representative of inward (blue) or outward (green) currents. Right: molecular components of each ionic current, separated in channel-forming subunits, auxiliary subunits and interacting proteins (adapted from: George A.L. Jr. [5]).

**Figure 3 biomedicines-10-00639-f003:**
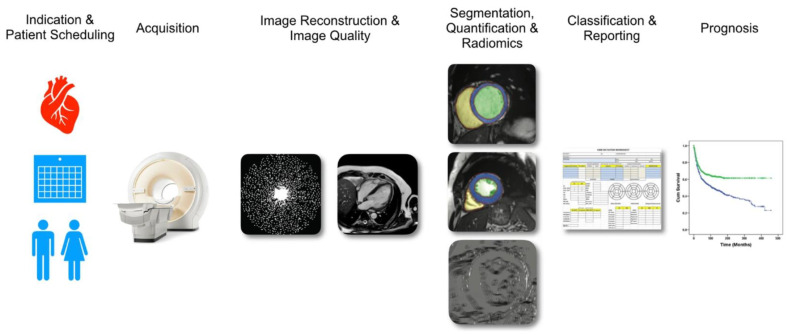
Aspects of cardiovascular magnetic resonance imaging that could be positively impacted by machine learning. These aspects range from patient scheduling to acquisition, image reconstruction, image segmentation, radiomic, classification and prognosis (adapted from Leiner et al. [70]).

**Figure 4 biomedicines-10-00639-f004:**
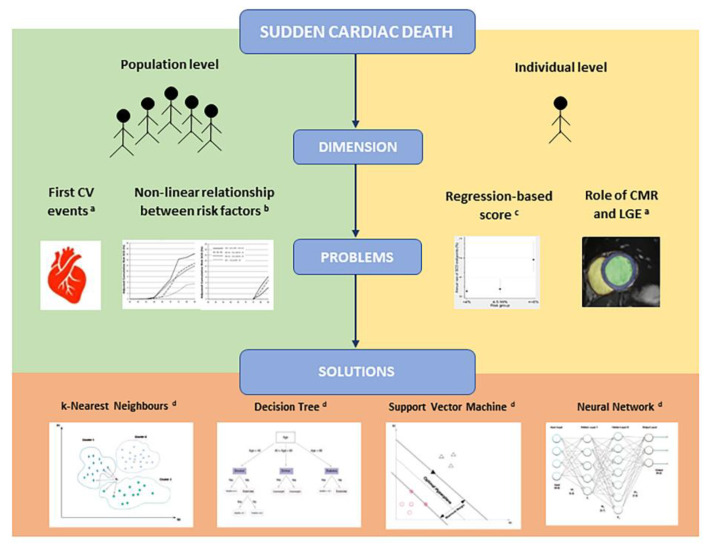
The figure retraces the structures of review. Sudden cardiac death (SCD) has been addressed in the two different dimensions of population level and individual level. Current evidence reveals many problems in the accuracy of risk stratification strategies. Solutions proposed in this review suggest the use of ML models to improve patient selection for an implantable cardioverter defibrillator for primary prevention. (a) Adapted from Leiner et al. [64]; (b) lifetime risk for SCD in male population at index age 45 years (left image) and 75 years (right image), adapted from Bogle et al. [6]; (c) annual rate of SCD end point within 5 years stratified according to HCM Risk-SCD), adapted from O’Mahony et al. [50]; (d) adapted from Al’Aref et al. [60].

**Table 1 biomedicines-10-00639-t001:** Comparison of studies that investigated the utility of machine learning models for CVD prevention at a general population-based level.

First Author, Year	N° of Patients	Follow-Up	ML Algorithm	Performance Evaluation of ML-Model (AUC)	Comparing Model	Performance Evaluation of Comparing Model (AUC)
Unnikrishnan [27], 2016	2406		SVM	0.71	Framingham	0.57
Weng [21], 2017	378,256	10 y	LR RF GBM NN	0.74 0.76 0.76 0.76	ACC/AHA 2013	0.73
Zarkogianni [28], 2017	560	5 y	NN LR	0.71 0.55	-	-
Kim [29], 2017	4244	10 y	NN NB LR SVM RF	0.79 0.74 0.72 0.50 0.70	-	-
Kakadiaris [30], 2018	6459	13 y	SVM	0.94	ACC/AHA 2013	0.72
Quesada [31], 2019	38,527	4 y	QDA NB NN ADA LDA LR (other 10 models)	0.71 0.71 0.70 0.70 0.70 0.70	REGICOR SCORE	0.66 0.63
Alaa [32], 2019	423,604	7 y	SVM RF NN ADA GBM	0.71 0.73 0.75 0.76 0.77	Framingham	0.72
Yang [33], 2020	29,930	3 y	NB BT ADA RF	0.71 0.75 0.79 0.79	Framingham	0.76
Li [34], 2020	3,661,932	10 y	Logistic methods, RF, NN, GBM and parametric models with different software package		Framingham QRISK3	0.86 0.88

ADA: AdaBoost; AUC: area under the Receiver Operating Characteristics curve; BT: bagged trees; GBM: gradient boosting machines; LDA: linear discriminant analysis; LR: logistic regression; ML: machine learning; RF: random forest; SVM: support vector machine; NB: naïve bayes; NN: neural network; QDA: quadratic discriminant analysis; y: year.

**Table 2 biomedicines-10-00639-t002:** Operation of algorithms used in the main machine learning studies in cardiovascular disease prevention.

Algorithm	Operation
AdaBoost	Generates a sequence of weak classifiers, where at each iteration, the algorithm finds the best classifier based on the current sample weights. Samples that were incorrectly classified in the kth iteration receive more weight in the (k + 1)st iteration, while samples that are correctly classified receive less weight in the subsequent iteration. At each iteration, a stage weight is computed based on the error rate at that iteration. The overall sequence of weighted classifiers is combined into an ensemble and has a strong potential to classify better than any of the individual classifiers.
Naïve Bayes Classification	It is a simple probabilistic classification method based on Bayes’ theorem with the “naive” assumption of conditional independence.
Bagged trees	Extracts multiple random datasets to fit multiple decision tree models in order to improve the models’ performance. Each decision tree differs because of the subset data, and the final prediction results are determined based on the prediction of all trees.
Linear discriminant analysis Quadratic discriminant analysis	Both use the maximum-likelihood framework to classify data by adding the assumption that data from each condition has a multivariate normal distribution. This assumption allows the likelihood of any input to be computed quickly with a closed-form probability density function for the multivariate normal.
Support Vector Machine	The classifier is constructed by projecting training data into a higher dimensional space via mappings known as kernels, and devising in this new space a boundary (formally known as a hyperplane), which maximizes separation between the classes. New examples are then projected into this higher dimensional space, where this previously learned boundary is used to assign labels.
K-nearest neighbor	Every object being classified is compared to its k nearest training examples via a distance function, where k is an integer; its label is then assigned by majority vote.

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
