# Peer review of "Strategies for Sudden Cardiac Death Prevention"

_biomedicines, 2022, doi:10.3390/biomedicines10030639_

Round 1

Reviewer 1 Report

The paper of Coriano and Tona is an interesting review of a possible future approach to the risk stratification of SCD. The topic is of interest as nowadays there is a lot of confusion  in the identification of people possible at risk, mainly becouse so often there is not a gold standard even for the true diagnosis of the disease, and the different weight and importance of various risk factors are not well known

If we send input wrong variables in a data base, the results will never give a relevant answer. If we are 100% sure that a pathopysiological risk factor is the responsible for SCD than the risk stratification is so easy, but if we lack this knowledge and rely on incomplete, popular,  but no evidence based  data (including guidelines), the under or over extimation of this input factor  will never produce true evidence based risk stratification data.

In  inheredited conditions many genetic abnormalities have been identified (Sudden Cardiac Death—A New Insight Into Potentially Fatal Genetic Markers Dragan Primorac 2021) but so often these genetic abnormalities do not correspond to a typical fenotype (and viceversa). For instance in the brugada syndrome SCN5A abnormalities are found in not more than 30% of subjects and in the same family sometime the fenotype does not fit with the genotype and viceversa (. Probst V, Wilde AM, Barc J. SCN5A mutations and the role of genetic background in the pathophysiology of Brugada syndrome. Circ Cardiovasc Genet 2009;2:552–557). The same may happens in LQTS, in Polymorfic ventricular tachycardia, and in the cardiomyopathies which means that these genes cannot surely to be considered the gold standard for the diagnosis and risk stratification even with an artificial intelligence.  We are still so ignorant on the pathophysiology of these diseases as other unknow genes may be involved and different structural and abnormalities might be the true origin of the electrophysiologica substrate as proposed for these entities more than three decades ago, but not well investigated yet. At present time in my opinion, the input of these genetic  variables is surely  to be considered in the risk stratification performed both by human and artificial intelligence, but we must be so cautious with the weight of their importance.

Concluding, in my opinion this review is important as it well illustrates the future methodological approach of risk stratification in different heart disease, but it must be clearly stated that the high relevance of the final results depend by the evidence based importance of the input variables, wich at this time is still a relevant problem

Author Response

It must be clearly stated that the high relevance of the final result depends by the evidence based importance of the input variables, which at this time is still a relevant problem.

We thank the Reviewer for the advice. We revised the manuscript and added a comment that highlighted the importance to use variable with a proven correlation with the event that our algorithm would predict. (Highlighted in gray)

----------------------------------------------------------------------------------------------------------------------------------------------------------------

Reviewer 2 Report

In the review "Strategies for sudden cardiac death prevention" the authors summarize the knowledge concerning  SCD prevention at population and individual level as well the role of mavhine learning for risk stratification 

The paper could be of interest however some concerns should be adressed

  1. Please revise the manuscprit in order to improve readibility and correct typos and
  2. In the paragraph "Sudden cardiac death risk management at individual level: current recommendations, gap in evidence and new perspective" some sentence regarding risk stratification in cardiomyopathies according to American Guidelines should be added. 
  3. In the same  paragraph add  a subsection on channelopathies 
  4. Add a pargraph on future prospective 

Author Response

Please revise the manuscprit in order to improve readability and correct typos.

We thank the Reviewer for the advice. We revised the manuscript and corrected typos. (Highlighted in turquoise)

----------------------------------------------------------------------------------------------------------------------------------------------------------------

In the paragraph "Sudden cardiac death risk management at individual level: current recommendations, gap in evidence and new perspective" some sentence regarding risk stratification in cardiomyopathies according to American Guidelines should be added.

We thank the Reviewer for the comment. We added a sentence regarding risk stratification in nonischemic dilated cardiomyopathy and arrhythmogenic cardiomyopathy according to American guidelines. (Highlighted in yellow)

----------------------------------------------------------------------------------------------------------------------------------------------------------------

In the same paragraph add a subsection on channelopathies.

We thank the Reviewer for the suggestion. We added a paragraph about current strategies for risk stratification in channelopathies. (highlighted in green)
----------------------------------------------------------------------------------------------------------------------------------------------------------------

Add a pargraph on future prospective.

We thank the Reviewer for the suggestion. We added a paragraph that highlights future perspective. (highlighted in red)
----------------------------------------------------------------------------------------------------------------------------------------------------------------

Round 2

Reviewer 2 Report

The authors adressed the majority of reviewer concerns.

As minor point add some sentence on HCM risk stratification according to AHA/ACC guidelines.

Author Response

We  have revised accordingly.